# Passive Smokers’ Support for Stronger Tobacco Control in Indonesia

**DOI:** 10.3390/ijerph17061942

**Published:** 2020-03-16

**Authors:** Renny Nurhasana, Suci Puspita Ratih, Komara Djaja, Risky Kusuma Hartono, Teguh Dartanto

**Affiliations:** 1Urban Studies Program, School of Strategic and Global Studies, Universitas Indonesia, Jakarta 10430, Indonesia; rennynurhasana@ui.ac.id (R.N.); komara.djaya@ui.ac.id (K.D.); 2Center for Social Security Studies, Universitas Indonesia, Jakarta 10320, Indonesia; 3Department of Public Health, Faculty of Sports Science, Universitas Negeri Malang, Malang 65145, Indonesia; 4Department of Public Health, Sekolah Tinggi Ilmu Kesehatan Indonesia Maju, Jakarta 12630, Indonesia; risky.kusuma@stikim.ac.id; 5Department of Economics, Faculty of Economics and Business, Universitas Indonesia, Depok 16424, Indonesia; teguh.dartanto@ui.ac.id

**Keywords:** passive smoker, tobacco control, regulation, tax increase

## Abstract

Secondhand smoke exposure in Indonesia is high, especially compared to other Southeast Asian countries. Passive smoking leads to negative impacts on health and socio-economic well-being. Therefore, increasing the price of cigarettes and, thereby, increasing barriers to access to cigarettes could be an effective way to reduce smoking prevalence and protect people from second-hand smoke. This study aims to assess passive smokers’ support for cigarette price increases in Indonesia. We perform a quantitative analysis with a cross-sectional design. The data were obtained through phone-based interviews of 1000 respondents aged 18 and older in Indonesia. Only 596 nonsmokers were included to be further analyzed in this study. This study found that 44.1% respondents have at least one family member who smokes. We considered the respondents’ age, gender, education level, employment, and the number of people living in the respondent’s household that are exposed to passive smoking. Our results demonstrate that passive smokers support stronger tobacco control such as increasing cigarette prices, regulating smoking behavior using a religious approach (*Fatwa*), and applying more effective pictorial health warnings.

## 1. Introduction

Tobacco use is the leading contributor to non-communicable disease and kills an estimated seven million people in the world each year. Moreover, an estimated 890,000 non-smokers died per year as a result of exposure to secondhand smoke (passive smokers) [1]. The data from the Indonesia National Socioeconomic Survey and the Indonesia Basic Health Research show that smoking prevalence in Indonesia remains high (rose from 27.0% in 1995 to 36.3% in 2013) [2,3]. Although recent data showed a slight decrease in the prevalence of tobacco consumption to 33.8% in 2018, the prevalence is still the highest in Southeast Asia, as reported by Southeast Asia Tobacco Control Alliance (SEATCA) [3,4]. Smoking prevalence is 19.1% in Thailand, 22.8% in Malaysia, 12% in Singapore, 26.1% in Myanmar, 27.9% in Lao PDR, 19.9% in Brunei Darussalam, 22.5% in Vietnam, 28.3% in The Philippines, and 16.9% in Cambodia [1]. 

In Indonesia, about 60% of youth are exposed to smoke outside the home and 57% are exposed to smoke at home [1]. Around 85% people are exposed to secondhand smoke inside restaurants, while 78% people are exposed to that at home [1]. Public transportation, universities, educational facilities, and health care facilities are supposed to be smoke-free in some cities/municipalities; however, the enforcement of comprehensive smoke-free laws in Indonesia is one of the weakest among Asian countries. The implementation of the regulation mostly depends on the local governments. Therefore, it is quite difficult to legislate against smoking in public places in the national context. Moreover, the smoke-free policy is applied at few universities. Among those stating that they have applied the policy, the enforcement is weak at best. Indonesia is known to be one of countries that has not ratified the WHO Framework Convention on Tobacco Control (FCTC). It is clearly stated in the Article 8 of the FCTC that the countries are urged to implement comprehensive smoke free policies. 

The term passive smoker commonly refers to those who breathe other people’s smoke. Passive smokers have a high risk of smoking-related non-communicable diseases. The literature shows that exposure to tobacco smoke in the environment contributes to athero-thrombosis, which leads to cardiovascular disease-related morbidity and mortality [5]. Moreover, passive tobacco exposure also contributes to cancer risk, particularly lung and colorectal cancers among men and women and breast and cervix cancers among women [6]. Adolescents with smoking parents or guardians are known to have higher likelihood of being a passive smoker, regardless their own smoking status [7]. Moreover, these adolescents may begin smoking after being exposed to their parents’ smoking behaviour, further amplifying the negative health impacts.

Smoking behavior not only affects one’s health but also one’s socio-economic successes. Exposure to smoking is correlated with increased stunting and hospitalization of children [8]. The study also found that high numbers of cigarettes consumed per day is negatively correlated with expenditure on nutritious food [8]. Thus, smoking is a risk factor for poor quality of life. Stronger tobacco control is critical to protect society, especially children and youth, from the harms of smoking behavior. Our study aims to assess the passive smokers’ support for stronger tobacco control in Indonesia.

## 2. Methods

The present analysis is a quantitative study with a cross-sectional design using survey method. The data were obtained through phone-based interviews of 1000 respondents age 18 and over in Indonesia. Of those 1000 respondents, only non-smokers (never-smoker and ex-smoker) were included in further data analysis. In total, we had data for 596 non-smoker respondents. In this study, we identified passive smokers as those whose family members were smokers. The respondents were identified based on a complete list of registered mobile phone users in Indonesia for different network providers: Telkomsel, Indosat, XL, Tri, SmartFren, and others. The data from The Ministry of Communication revealed that the total registered users of mobile phone number in Indonesia in 2018 were 296,270,269. Therefore, this study used systematic random sampling to generate samples that represented each provider. The first phone numbers contacted were chosen randomly from each provider. Afterwards, the next phone numbers were chosen with an interval between 100,000 and 200,000 from one number to another. The total users of network providers and the number of samples generated from the percentage of total users can be seen in Table 1. The respondents in this study comprised almost all provinces in Indonesia (91.2%). 

Ethical approval from an academic institution was obtained prior to the data collection. Besindes, oral consent was obtained from each respondent before starting the interview, which included permission to record the conversation. The respondents covered almost all provinces in Indonesia. The interviews were conducted from 1 May, 2018 to 31 May, 2018. The variables measured in this study included demographic characteristics such as ages, gender, education level, income, occupation, area of living, and number of persons living in their houses. This study also assessed smoking status that was measured by smoking activity in the last 30 days. Moreover, smoking behavior of respondent’s family members was also evaluated by asking wheter or not there was any smoker in their houses. The attitudes toward smoking behavior as well as tobacco control efforts were also measured by asking respondent’s perceptions about smoking behavior in general and of their family members, increase in cigarette price, prohibition of smoking in Islam, and pictorial health warnings applied on cigarette packs. This study performed a descriptive analysis and a multivariate logistic regression model. 

## 3. Results

### 3.1. Demographic Characteristics of Passive Smokers

Amongst the 569 non-smoker respondents, 44.1% (n = 263) of them reported to have at least one family member that smokes (passive smoker), while 55.9% reported that they did not have any smoking family members. Among passive smokers (n = 263), 39.0% identified themselves as ex-smoker (OR 0.8, 95% CI 0.5–1.4, *p* value 0.391)). Moreover, of passive smokers in this study, 39.9% were aged 25–38 years old (OR 1.8, 95% CI 1.2–2.7, *p* value 0.008) while 40.3% were aged ≥39 (OR 2.1, 95% CI 1.3–3.5, *p* value 0.005). In terms of respondents’ gender, most are female (OR 0.4, 95%, CI 0.3–0.6, *p* value < 0.001)). Both age and gender were found to be significantly associated with passive smoking (see Table 2). 

Additionally, of respondents with a low education and those that were unemployed, the majority were passive smokers (58.33% of respondents with a low education and 50.88% of unemployed respondents, respectively). Of the respondents with a high education level, 35.9% of them were exposed to secondhand smoke (OR 2.8, 95% CI 1.1–7.2, *p* value 0.036). Among our participants, the proportion of respondents with exposure to passive smoking was similar for those who lived in district (urban) and those that lived in municipality (rural) areas (43.6% and 45.1% respectively). Furthermore, passive smokers were also found mostly in houses that are inhabited by more than five people (59.80%) and number of people in the household was found to be significantly associated with passive smoking (OR 0.4, 95% CI 0.2–0.7, *p* value 0.001).

### 3.2. Attitude towards Smoking Behavior 

Based on univariate analysis, this study revealed that those who were exposed to cigarette smoking at home (n = 263) have smoking fathers and husbands (32.7% and 28.5% respectively). Moreover, smoking behaviors of older brothers (18.3%) and little brothers (6.5%) were also found to be the sources of exposure to secondhand-smoke at home (see Table 3). Among passive smokers, most (99.2%) acknowlegded that smoking is harmful. Almost all (98.1%) were fully aware of the harmful effects of smoking. Moreover, 93.16% of respondents want to encourage their family members to quit smoking, but admitted that it was difficult to encourage those who are active smokers to quit smoking. 

### 3.3. Support towards Stronger Tobacco Control

This study found that 86.7% of passive smokers (n = 263) supported the rise in the price of cigarettes to prevent underage buyers from acquiring packs of cigarettes. Moreover, 86.69% of passive smokers agreed to the increase in cigarette price because they believed it could help their family members to quit smoking. Increasing tobacco product tax excises affects the price of cigarettes, making them less affordable for children and the poor. The excise tax increase is known to have a significant effect on the reduction of smoking prevalence and the number of smoking-attributable deaths. 

The same study also found that 65.94% agreed that there should be a Fatwa saying that smoking is haram (prohibited in Islam). Fatwa is a formal ruling in Islam given by a qualified Islamic scholar (Ulama), in response to questions from the people and the Islamic court. In Indonesia, Fatwas are shared by the Indonesian Council of Ulama (MUI) or other Islamic organisations. 

Another effort to reduce the use of tobacco that has been implemented in Indonesia is the use of Pictorial Health Warnings (PHWs) on tobacco packs. The majority of passive smokers in the present study believed that the current pictorial health warning printed on cigarette packs in Indonesia were not effective enough (50.57%). Only 13.26% of respondents stated that PHWs could convince people to quit smoking, and around 36.56% said that it had the potential to raise awareness about the harms of cigarettes (see Table 4).

## 4. Discussion

### 4.1. Demographic Characteristics of Passive Smokers

This study found that passive smokers, most were in the younger age (18–24 years old) and female. Both age and gender were found to be significantly associated with passive smoking. This finding is in line with the study on health risk factors in adolescents in India (n = 400) in which age is strongly associated with passive smoking [9]. Additionally, a study in China (n = 9788 non-smoking women) showed that the prevalence of passive smoking among non-smoking women in Jilin Province was 60.6% and 42.9% reported that they were exposed to cigarette smoking at home [10]. A cross-sectional study from Gambia (n = 10,392 students) showed that exposure to second-hand smoke was more common in girls than boys [11].

On the contrary, based on data by the WHO, more men are exposed to passive smoking at home than women in Indonesia (81.4% and 75.4% respectively). Similarly, the data also revealed that Indonesian boys are more likely to be exposed to passive smoking than girls (61.7% and 51.7% respectively) [12]. That said, passive smoking is known to be a higher risk factor to stillbirths (14% in Indonesia) than active smoking behavior [13]. Hence, protecting women and girls from passive smoking is critical, especially women of childbearing age, and can be a key strategy to improve the health of women and children. 

Exposure to passive smoking in this study was similar for those who lived in district (urban) and those that lived in municipality (rural). A previous study indicated that urban households have lower exposure to passive smoking [13]. Likewise, a study on secondhand smoke exposure from 26 countries revealed that passive smoking is higher in urban areas relative to rural ones [14]. Furthermore, passive smokers were also found mostly in houses that are inhabited by more than five people (59.80%) and number of people in the household was found to be significantly associated with passive smoking (*p* value < 0.01). Densely populated houses are likely to be found in poorer households in which smoking prevalence is high.

### 4.2. Attitude towards Smoking Behavior 

Participants of this study mostly report that it is difficult to encourage their family members to stop smoking although they are aware of the harms of cigarettes. Our previous study revealed that smokers are likely aware of the harms of smoking; however, because cigarettes contain the addictive substance nicotine, smoking cessation is difficult. In this case, the help and support of family members are essential for those who are trying to end smoking [13]. Spreading awareness about the side effects of passive smoking would teach people about the danger of smoking and enable them to make informed decisions about their health and the health of their families [15]. A randomized control trial among 1158 families with fathers that smoke daily and mothers that do not smoke was conducted in Hong Kong to examine the effectiveness of a family-based intervention to help smoking cessation [16]. The data showed that the family-based intervention resulted in a greater prevalence of 7-day and 6-month self-reported abstinence, demonstrating that the support of family members was effective in increasing smoking abstinence among fathers in the study. 

### 4.3. Support towards Stronger Tobacco Control

Regarding stronger tobacco control regulations, the respondents mostly supported the rise in the price of cigarettes to prevent underage buyers from acquiring packs of cigarettes. They also agreed the increase in cigarette price, because they believed it could help their family members to quit smoking. Raising cigarette taxes is one of the most effective tobacco control policies [17,18]. Increased taxes encourage smokers to either reduce the amount they smoke or quit smoking completely; they also discourage young people from starting [17,18]. However, a comprehensive effort should be conducted such as educating people about the harms of cigarettes, stronger smoke-free regulations, approaching local governments and religious and/or community leaders, and providing more adequate cessation services. 

Indonesian governments have raised cigarette excise tax by 23% and set a minimum selling price to reach an average of 35% per 1 January 2020. Increasing tobacco product tax excises affects the price of cigarettes, making them less affordable for children and the poor. The excise tax increase is known to have a significant effect on the reduction of smoking prevalence and the number of smoking-attributable deaths. A simulation study in low- and middle-income countries in the Asia-Pacific Region was conducted and showed that, given an average annual cigarette price increase of 9.5%, the average annual cigarette consumption would decrease by 3.6% [19]. On the other hand, the average annual tobacco tax revenue would increase by 16.2%. Additionally, the number of averted smoking-attributable deaths (SADs) would be highest in China, followed by Indonesia and India. In total, the simulation study found that there would be over 17.96 million lives saved by an increase in cigarette taxes [19].

Increasing cigarette prices is a practical way to help improve population health and quality of life and to allow people to save money on health expenses. One study from Gambia reported that increasing cigarette prices reduced tobacco consumption and generated significant government revenue from tobacco products [20]. Additionally, in Gambia, increasing tax rates on tobacco products reduced the substitution between tobacco products [20]. 

As a country with the largest Muslim population in the world, religion-based interventions could be important in order to persuade people to use less tobacco in Indonesia. Majority of the respondents in this study agreed that there should be a Fatwa saying that smoking is haram (prohibited in Islam). Fatwa is a formal ruling in Islam given by a qualified Islamic scholar (Ulama), in response to questions from the people and the Islamic court. In Indonesia, Fatwas are shared by the Indonesian Council of Ulama (MUI) or other Islamic organisations. In 2009, the clerics of Islamic scholars agreed to issue a Fatwa on the smoking law, stating that smoking should be avoided and prohibited. Nevertheless, the statement said that cigarettes were prohibited only for children, pregnant women, and smoking in public places. The clerics requested that the government and the House of Representatives agree to it and acknowledge it as part of the law. Besides that, the agreement also urged the government to switch tobacco for other products, rather than cigarettes. 

Other Islamic countries such as the United Arab Emirates and Egypt have prohibited smoking in public areas. They have also posted anti-smoking advertisements, promotions, and sponsors both in print and electronic media, which contributed to reductions in the amount of tobacco consumed in the countries. Thus, tobacco control efforts can be conducted using various approaches, including through religious institutions. Issuing tobacco Fatwa stating that smoking is haram in Islam could be considered in conjunction with increasing cigarette prices to decrease tobacco consumption. Therefore, religious leaders and the government, especially the Ministry of Religion Affairs, should take action and conduct efforts to reduce smoking prevalence in Indonesia, where most of the population is Muslims. 

Indonesian governments have also implemented another effort to reduce the use of tobacco that by applying Pictorial Health Warnings (PHWs) on tobacco packs. The regulation requires text warning saying “Smoking kills you” and graphical warnings about the harms of cigarettes on the top of the front and back sides of the packs. The PHWs must cover 40% of each of the main faces of the cigarette packs. The majority of passive smokers in the present study believed that the current pictorial health warning printed on cigarette packs in Indonesia were not effective enough to convince people to quit smoking. The respondents also perceived PHWs as less effective to raise awareness about the harms of cigarettes. These findings suggest that passive smokers assume that PHWs should be more frightening when it comes to educating the public on the danger of smoking. The size of current PHWs in Indonesia is 40% which is still relatively small when compared to other Asian countries, such as Thailand (85%), India (85%), Nepal (90%), and Timor Leste (92.5%). A study by Tobacco Control Support Center in 2017 found that the current PHWs is perceived as a little bit scary (36.0%) [21]. When the size of PHWs was enlarged to 75% and 90%, the number of respondents who perceived PHWs as extremely scary increased to 21.3% and 456.5% respectively [21]. These findings indicate that the larger the size of PHWs, the more effective they are in depicting the negative impact of smoking. The use of PHWs is perceived to be more compelling than text-only warnings, according to a literature review of studies in Asian countries [22]. Thus, a larger size and renewal of the pictures should be improved for stronger tobacco control in Indonesia.

## 5. Conclusions

Based on the study and discussion, we conclude that the prevalence of passive smoking is 44.13% among 569 nonsmokers in this study which was identified by the exposure to cigarette smoke at home. Exposure to cigarettes at home is most common among younger and female responents, respondents with lower education level, unemployed/students, and respondent who live with more than five people in the house. Most of passive smokers (93.16%) want to encourage smokers in their family to quit. In addition, the majority of passive smokers support stronger tobacco control, such as increasing cigarette price, applying a religious apporach (Fatwa) to convince people that smoking is bad, and applying more effective pictorial health warnings. Thus, this study suggests that the government should apply stronger tobacco control efforts to protect nonsmokers, especially women and children, against the harms of cigarettes.

## Declaration 

### Consent to Participate 

In this study, we received consent to use the personal information of each participant before the interview. 

### Consent for Publication 

Not applicable 

### Availability of Data and Materials 

All data analyzed during this study are included in this published article. 

### Competing Interests 

All authors declare that they have no conflict of interests. None of the authors benefited from any direct or indirect funding from the tobacco industry or any other company. 

## Figures and Tables

**Table 1 ijerph-17-01942-t001:** Population and sample of the study.

Classification	Total Population	Percentage	Sample	Final Sample
Wireless Network Provider				
Telkomsel	139,934,665	47.2%	472	473
Indosat	100,538,309	33.9%	339	339
XL	42,362,769	14.3%	143	142
Tri	12,782,993	4.3%	43	42
Smartfren	588,086	0.2%	2	0
Others	63,447	0.0%	0	0
TOTAL	296,270,269	100.0%	1000	1000
Region				
Western Indonesia	207,702,000	79.3%	793	803
Central Indonesia	42,126,000	16.0%	160	170
Eastern Indonesia	12,062,900	4.7%	47	17
TOTAL	261,890,900	100.000%	1000	1000

**Table 2 ijerph-17-01942-t002:** Demographic characteristics and smoking status of respondents.

Characteristics	Measurement	Passive Smoker				Odds Ratio (95% CI)	*p* Value
		Yes		No			
		n	%	n	%		
Smoking Status	Never-smoke*	226	45.1	275	54.9		
	Ex-smoker	37	39.0	58	61.1	0.8 (0.5–1.4)	0.391
Age	18–24*	94	54.0	80	46.0		
	25–38	115	39.9	172	60.1	1.8 (1.2–2.7)	0.008
	≥39	54	40.3	80	59.7	2.1 (1.3–3.5)	0.005
Gender	Male*	79	32.2	166	67.8		
	Female	184	52.4	167	47.6	0.4 (0.3–0.6)	0.000
Level of Education	Low*	14	58.3	10	41.7		
	Middle	137	52.7	133	47.3	1.4 (0.5–3.5)	0.496
	High	112	35.9	200	64.1	2.8 (1.1–7.2)	0.036
Income (million rupiah)	<2.9*	85	44.7	105	55.3		
	3–6.9	122	48.0	132	52.0	0.6 (0.4–0.9)	0.013
	>7	56	36.8	96	63.2	0.9 (0.5–1.6)	0.753
Occupation	Unemployed/student*	87	50.9	84	49.1		
	Self-employed	34	37.4	57	62.6	1.0 (0.6–1.8)	0.977
	Employee (Private and Gov’t sectors)	94	39.2	146	60.8	0.9 (0.6–1.5)	0.721
	Laborer	48	51.1	46	48.9	0.6 (0.3–1.1)	0.093
Area of Living	Municipality*	88	45.1	107	54.9		
	District	175	43.6	226	56.4	1.1 (0.7–1.6)	0.699
Number of Persons in the House	<3 persons*	71	37.2	120	62.8		
	4–5 persons	131	43.2	172	56.8	0.8 (0.5–1.1)	0.153
	>5 persons	61	59.8	41	40.2	0.4 (0.2–0.7)	0.001

* referent; n = 263.

**Table 3 ijerph-17-01942-t003:** Attitude towards smoking behavior.

Variables	n	%
Family members who smoke		
Father	86	32.7
Husband	75	28.5
Older brother	48	18.3
Little brother	17	6.5
Kids	16	6.1
Mother	13	4.9
Other	8	3.0
Agreed that smoking is harmful	261	99.2
Aware of the harms of Cigarettes	258	98.1
Want to encourage family members to stop smoking	245	93.2

**Table 4 ijerph-17-01942-t004:** Support towards stronger tobacco control.

Tobacco Control Efforts	n (263)	%
**Increase in cigarette price**		
Agreed	228	86.7
Disagreed	35	13.3
**Fatwa that prohibits smoking in Islamic view**		
Agreed	172	65.8
Disagreed	90	34.2
**Pictorial Health Warnings**		
Current PHWs are not effective	133	50.6
Current PHWs can convince people to quite smoking	32	12.2
Current PHWs have potential to raise awareness	98	37.3

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
