# Peer review of "Passive Smokers’ Support for Stronger Tobacco Control in Indonesia"

_ijerph, 2020, doi:10.3390/ijerph17061942_

Round 1

Reviewer 1 Report

My main comment concerns the simplistic conclusions drawn from the research. In particular, this statement: “Raising taxes is … a win-win solution for society and the government.” This statement implies that there are no potential downsides to raising excise taxes, which simply is not true. It is known that raising cigarette taxes has these effects: 1) tax revenue increases, 2) cigarette consumption decreases, and 3) other things equal, there will be more illicit trade in cigarettes. See, for example, Prieger and Kulick, “Cigarette taxes and illicit trade in Europe”, Economic Inquiry Volume 56, Issue3 (2018), 1706-1723, the special issue on the illicit tobacco market in Trends in Organized Crime (December 2016, Volume 19, Issue 3–4) and the many sources cited in these articles. For Indonesia in particular, one could reference Suprihanti et al., “The Impact of Cigarette Excise Tax Policy on Tobacco Market and Clove Market in Indonesia” International Journal of Economics and Financial Issues, 2018, 8(6), 54-60, which concluded that a consequence of a higher tax on tobacco “was the number of smuggled cigarettes raised”. The third point is obvious to economists and well documented in their literature but often ignored or dismissed without reason by tobacco control researchers. To be clear, my quibble is not with the conclusion that raising taxes might have salutary effects on health and state budgets, but rather that it is presented as a simplistic decision to make. Regardless of whether the authors think tobacco control through higher taxes is a good thing, evidence that is perhaps to the contrary should not be ignored. To be clear: policymakers must plan for fighting this potential unintended consequence of raising taxes.

The opening statements are typical of those in the tobacco control literature, but are stronger than warranted by the literature upon which they are based. The statements are: “Tobacco use kills more than seven million people in the world each year. Moreover, 890,000 non-smokers died per year as a result of exposure to secondhand smoke (passive smokers).” The proper statement must have qualifiers such as “an estimated seven million people” etc. Since these figures are based on statistical estimates of unknown population quantities, they all have standard errors and confidence intervals (which somehow never seem to be reported in the final death tolls). A figure such as “890,000” is therefore an estimate, and must be presented as such. These two sentences can be combined: “Prevalence of cigarette smoking in Indonesia is high. The data from the Indonesia National Socioeconomic Survey and the Indonesia Basic Health Research show that smoking prevalence in Indonesia remains high.” This statement appears to be nonsensical as written, since the totals should add up to 100%: “Around 85% of secondhand smoke occurred inside restaurants while 78% occurred at home.” What I think the authors are trying to say is that 85% of people (or people reporting some exposure) get exposure from restaurants, and 78% of them get it at home. This needs to be rewritten to clarify. “Public transportation, universities, educational facilities, and health care facilities are supposed to be smoke-free” is vague. Be clear that meaning is: illegality. “In households with daily-smoking parent, high numbers of cigarettes consumed per day can contribute to lower expenditure on nutritious food.” I cannot see the source cited, because it isn’t published, but I suspect that the correct statement would be that high numbers of cigarettes consumed per day is negatively correlated with expenditure on nutritious food. I really doubt that the source has a convincing study design to determine a causal relationship, and if not, careful language must be used. “We were unable to include passive smokers that are exposed to cigarette smoke at the workplace or other public places.” A statement on why, perhaps in a footnote, is appropriate. Seems like a simple enough question to ask. Given the limitation of the sample to mobile phone users, a statement on how the subpopulation of mobile phone users in the country compares to the overall adult population would be useful. And what proportion of adults use a mobile phone? “Those that consider themselves smokers agree that smoking is bad for one’s health…” This is a puzzling statement, since the authors said that the study sample consisted only of passive smokers. So what does this mean? They consider themselves to be passive smokers? This should be clarified. “Moreover, 86.69% of passive smokers agreed to the increase in price, because they believed it could help their family members to quit smoking.” The meaning of this sentence differs whether the second comma is included or not. I believe the authors intend the meaning given when that comma isn’t there. There is also no discussion in the paper of the fact that the work is not based on a probability sample, and the potential implications for computing standard errors, whether these estimates are unbiased for the population, etc.

Reviewer 2 Report

See my detailed comments in the attached pdf-file.

Round 2

Reviewer 2 Report

I agree with the revisions. However, I do not accept the statement connected to reference 17. Tobacco industry is exaggerating illicit trade. Also the statement in the manuscript is misrepresenting the content of the cited article.

Author Response

We have revised the sentence connected to reference 17, so it sounds more neutral.